# PP2A-B55 Holoenzyme Regulation and Cancer

**DOI:** 10.3390/biom10111586

**Published:** 2020-11-22

**Authors:** Perrine Goguet-Rubio, Priya Amin, Sushil Awal, Suzanne Vigneron, Sophie Charrasse, Francisca Mechali, Jean Claude Labbé, Thierry Lorca, Anna Castro

**Affiliations:** Centre de Recherche en Biologie Cellulaire de Montpellier (CRBM), Université de Montpellier, Équipe Labellisée “Ligue Nationale Contre le Cancer”, CNRS UMR 5237, 1919 Route de Mende, CEDEX 5, 34293 Montpellier, France; perrine.goguet@crbm.cnrs.fr (P.G.-R.); priya.amin@crbm.cnrs.fr (P.A.); sushil.awal@crbm.cnrs.fr (S.A.); suzanne.vigneron@crbm.cnrs.fr (S.V.); sophie.charrasse@crbm.cnrs.fr (S.C.); francisca.mechali@crbm.cnrs.fr (F.M.); jean-claude.labbe@crbm.cnrs.fr (J.C.L.)

**Keywords:** PP2A-B55, cancer, Greatwall, Arpp19, ENSA

## Abstract

Protein phosphorylation is a post-translational modification essential for the control of the activity of most enzymes in the cell. This protein modification results from a fine-tuned balance between kinases and phosphatases. PP2A is one of the major serine/threonine phosphatases that is involved in the control of a myriad of different signaling cascades. This enzyme, often misregulated in cancer, is considered a tumor suppressor. In this review, we will focus on PP2A-B55, a particular holoenzyme of the family of the PP2A phosphatases whose specific role in cancer development and progression has only recently been highlighted. The discovery of the Greatwall (Gwl)/Arpp19-ENSA cascade, a new pathway specifically controlling PP2A-B55 activity, has been shown to be frequently altered in cancer. Herein, we will review the current knowledge about the mechanisms controlling the formation and the regulation of the activity of this phosphatase and its misregulation in cancer.

## 1. Introduction

Most cellular processes are regulated by protein phosphorylation. This post-translational modification allows cells to react rapidly to the external environment by controlling the activity, localization, and substrate specificity of enzymes involved in most cell signaling pathways. Protein phosphorylation results from a fine-tuned balance between kinases and phosphatases, the disruption of which can lead to altered cellular behavior and many diseases including cancer. Despite their abundance, owing to conserved structural features, kinases have been the target of the cancer research field for decades. Deregulation of kinase activity has emerged as a major mechanism by which cancer cells evade normal physiological constraints on growth and survival. A considerable number of chemical inhibitors targeting these enzymes have been developed and have proven to be very successful in cancer therapy [1]. Nonetheless, these treatments have been limited by the development of drug resistance and/or loss of specificity [2]. 

Since they antagonize kinases, phosphatases are equally important for normal and tumoral cell proliferation. However, the role of these enzymes has been underestimated in the past, and phosphatases have been considered as housekeeping enzymes due to an erroneous preconceived hypothesis regarding their lack of specificity and regulation. Nevertheless, research over the last few years highlighted an essential role of phosphatase regulation in the control of cell proliferation and cancer development. These enzymes have now been identified as tumor suppressors involved in a large range of cancers [3] and have been considered as attractive targets for the development of new cancer therapies [4]. 

Phosphatases are classified into two different types, Protein Tyrosine Phosphatases (PTP) and serine/threonine phosphatases. The latter include three families: Phosphoprotein phosphatases (PPPs), metal-dependent protein phosphatases (PP2C), and Aspartate Based phosphatases (FCP1, SCP1) [5,6]. PPP represents the most abundant family of phosphatases in the cell and includes the members PP1, PP2A, PP2B, PP4, PP5, PP6, and PP7. 

This review will focus on PP2A and more specifically PP2A-B55. This holoenzyme, tightly controlled by the cell, has an essential function in cell division and cell growth. Although it has been involved in the control of a high number of physiological cascades, its specific role in cancer development and progression has only recently been highlighted thanks to the recent identification of the Greatwall (Gwl)/Arpp19-ENSA cascade, a new pathway specifically controlling PP2A-B55 activity. Increasing data identifies Gwl, Arpp19, and ENSA as proteins commonly misregulated in a high number of cancers displaying prognostic and therapeutic value. However, although their specific role in the control of the cell cycle has been well documented [7,8,9,10,11,12,13,14,15,16], very few reviews appeared in the literature on its involvement in cancer development and progression. This review will cover the different contributions that have recently appeared on the role of Gwl, Arpp19, ENSA, and PP2A-B55 in tumoral cascades and their misregulation in cancer. 

## 2. PP2A Structure 

PP2A is a family of holoenzymes formed of three different subunits: A catalytic subunit or C subunit, a scaffold subunit or A subunit, and a regulatory subunit or B subunit, and each of them comprises several isoforms. The combination of all these subunits results in the formation of more than 90 holoenzymes. This diversity creates specificity and confers to the phosphatase the capacity to distinctively recognize and dephosphorylate a high number of proteins. Additionally, the different isoforms display a particular tissue expression and a precise subcellular localization, conferring to every single phosphatase holoenzyme a unique pattern of activity for each specific substrate. 

The catalytic subunit is a globular protein ubiquitously expressed in almost all tissues that exists in two isoforms α and β. These two isoforms are encoded by two different genes (*PPP2CA* and *PPP2CB*) and display 97% of sequence similarity. However, despite this high similarity, these two isoforms appear to play a non-redundant role [17,18]. The C subunit can bind to the A subunit by its N-terminus to form a dimer, or to both the A and B subunits to form the active heterodimer enzyme. 

The scaffold A subunit is composed of 15 tandem Huntington-Elongation-A subunit-TOR (HEAT) repeats forming a horseshoe-like structure that binds to the different B subunits via its C-terminus, and to the catalytic C subunits by its N-terminus [19,20]. As for the C subunit, the scaffold A subunit displays isoforms α and β, sharing 86% of sequence similarity and encoded by *PPP2R1A* and *PPP2R1B* genes, respectively. 

The regulatory subunit or B subunit includes four different families: B (B55/PR55), B’ (B56/PR61), B’’ (PR48/PR72/PR130), and B’’’ (PR93/PR110/Striatins). There is a low sequence similarity between the four families, and each of them can display several isoforms (Figure 1A). 

The B/B55 family includes isoforms α, β, γ, and δ, encoded by the genes *PPP2R2A*, *PPP2R2B*, *PPP2R2C,* and *PPP2R2D.* These isoforms are differently distributed, with α and δ widely present in most tissues, whereas β and γ are mostly enriched in the brain. Expression is specific and developmentally regulated, with B55γ increasing and B55β decreasing after birth [21]. This family of proteins are structurally composed of seven-bladed β-propellers formed of WD40 repeats. B/B55 proteins make several contacts with the HEAT domains of the A subunit (repeats 1–7), but very few contacts with the catalytic subunit [22]. The association of C/A with B/B55 to form an active holoenzyme depends on the methylation of the catalytic subunit in L309 and is inhibited by the phosphorylation of this subunit on Thr304 and Y308 [23]. 

The B’/B56 family is formed by α, β, γ, δ, and ε isoforms (genes *PPP2R5A*, *PPP2R5B*, *PPP2R5C*, *PPP2R5D,* and *PPP2R5E*). They are structurally composed of eight HEAT-like repeats similar to the A scaffold subunits. Unlike the B subunits, the B’ regulatory subunits display quite a high number of contacts not only with the A subunit but also with the C subunit. 

The B’/B56 proteins can be directly modified by phosphorylation and nitrosilation, two post-translational modifications that severely impact the activity of the PP2A holoenzyme. Kinases such as MAPK or ERK can phosphorylate B56 and positively [24] or negatively [25,26] modulate PP2A-B56 activity. Conversely, the nitrosilation of these subunits induces the inhibition of the phosphatase [27].

The B’’/PR48 family contains isoforms α, β, and γ (genes *PPP2R3A*, *PPP2R3B,* and *PPP2R3C)*. All the members of this family are composed of Ca^2+^-binding EF motifs, and they require the presence of calcium to bind to A/C subunits [28]. They establish several contacts with the N-terminal Heat domains of the A subunit (1–7), and their C-terminus mediates the interaction with the C subunit and acts as a substrate binding site [29].

Finally, the B’’’/Striatin family includes the proteins Striatin, Striatin 3 (or SG2NA), and Striatin 4 (or Zinedin) (genes *STRN*, *STRN3*, and *STRN4)* [30]. These members cannot only bind to A/C subunits of PP2A, but can also interact with other proteins, including some kinases, to form the complexes named STRIPAKs [31]. These complexes participate to the modulation of several cell signaling cascades controlling cellular events such as clathrin-dependent endocytosis, cell junction stability, golgi assembly, polarity and migration, and Ca^2+^ signaling [32]. The association of the different components of these STRIPAK complexes are mediated by three different domains, a caveolin/Ca^2+^-CaM binding domain and a coiled-coil domain present at the N-terminus, and a WD-repeat domain at the C-terminus. It is proposed that these complexes could modulate the dephosphorylation of some of their components by the PP2A-associated phosphatase.

## 3. PP2A Activation

During PP2A active complex formation, the C subunit is submitted to different post-translational modifications and associations to ensure the correct and selective formation of the different PP2A holoenzymes. From its synthesis, the catalytic subunit is partly bound to α4 protein (Figure 1B). This binding keeps the C subunit inactive while facilitating its assembly with A and B subunits to form the trimeric complex. Moreover, it maintains C subunit levels by preventing its ubiquitination and degradation by the proteasome [33]. The assembly of the PP2A trimeric complex continues via the loading of Mg^2+^ metal ions into the catalytic site. This loading, which is crucial for acquisition of the Ser/Thr-specific phosphatase activity, requires ATP hydrolysis and is mediated by the phosphotyrosil phosphatase activator (PTPA) [34]. Once the Mg^2+^ ions are introduced to the catalytic site, the C subunit can bind to the A scaffolding subunit. However, the former can still be submitted to different posttranslational modifications such as phosphorylation on Y307 or carboxymethylation on L309. Reversible carboxymethylation is controlled by a balance between leucine carboximethyltransferase (LCMT-1) and the reverse enzyme PP2A-specific methylesterase 1 (PME-1). As described above, this post-translation modification of the catalytic subunit positively impacts the binding of B/B55 subunits while it does not affect the association of the rest of B subunits. Accordingly, deletion of the LCMT-1 gene on S Cerevisiae results in the inhibition of Cdc55 binding to yeast PP2A-C without affecting Tpd3p (A yeast subunit) or Rts1p (B56 yeast subunit) interaction [35,36,37,38]. In the same line, in mammalian cells, decreased methylation of PP2A C subunit resulting from LCMT-1 knockdown, prevents recruitment of B55 to PP2A C but not of B56 or PR48 [39,40,41]. Finally, when immunoprecipitated, methylated C subunit is exclusively present in endogenous PP2A-B55 holoenzyme [23]. L309 carboxymethylation is counterbalanced by the phosphorylation of C subunit on Y307. As such, phosphomimetic mutants of this residue cannot be methylated and fail to bind B55. 

## 4. PP2A Endogenous Inhibitors Regulating C Subunit and PP2A-B56

PP2A activity is modulated by several endogenous inhibitors. Although discovered almost twenty years ago [42], the mechanism of action and the specific PP2A holoenzymes targeted by these proteins have been only recently discovered. PP2A C and PP2A-B56 inhibitors include: ANP32a (or Inhibitor 1 of PP2A), SET (or Inhibitor 2 of PP2A), CIP2A, and Bod1 (Figure 1A).

ANP32a and SET were both isolated from bovine kidneys as two potent heat stable PP2A inhibitors [42]. Both proteins directly bind and inhibit the C subunit of PP2A independently of A and B subunits [43,44], and can be localized both in the nucleus and the cytoplasm. Their localization, as well as their binding to PP2A and their inhibitory activity, are modulated by phosphorylation [45,46,47]. SET stability is also controlled by its association to the SET binding protein 1 (SETBP1) [48]. ANP32a has been shown to modulate the PP2A-dependent dephosphorylation of the Tau [44] protein, whereas SET controls the dephosphorylation by this phosphatase of several substrates including Rec8 [49], ERK, AKT, c-Myc, Rb [50], PTEN [51], Mcl1 [52], and c-Jun [53]. SET can act as an oncogene, and its overexpression has been frequently detected and associated with poor prognostics in a series of cancers such as breast cancer, non-small cell lung cancer, or colorectal cancer [48,54,55]. Finally, increased mRNA levels of SET have also been associated with a poor prognosis in chronic lymphocytic leukemia [56].

CIP2A (Cancerous Inhibitor of PP2A), firstly identified as a co-precipitation partner of PP2A [57], is an inhibitor that specifically binds and inhibits the PP2A-B56 holoenzyme [58]. This inhibitor of PP2A-B56 has been shown to directly associate with and stabilize c-Myc by preventing its PP2A-B56-dependent dephosphorylation on S62 [57]. Similarly, CIP2A stabilizes E2F by preventing S364 PP2A-B56-dependent dephosphorylation and induces hyperactivation of Akt by inhibiting the dephosphorylation of S473 by this phosphatase [59,60]. CIP2A is widely overexpressed in human cancers including gastric, bladder, ovarian, tongue, hepatocellular, colon, non-small cell lung carcinoma, and chronic myelogenous leukemia, and is correlated with tumor grade [61].

As for CIP2A, Bod1 has been identified as a specific inhibitor of PP2A-B56 holoenzyme whose depletion results in changes in the phosphorylation balance at the kinetochores during mitosis, although the exact signaling pathway mediating this phenotype is unknown [62].

## 5. PP2A Endogenous Inhibitors Regulating PP2A-B55

Arpp19 and ENSA are the unique specific inhibitors of PP2A-B55 known so far [63,64]. Although PP2A-B55 is involved in the control of a high number of signaling cascades in the cell, due to their recent identification, little is known about the physiologic implication of these two inhibitors. Arpp19 and ENSA were first identified as two substrates of Gwl, a mitotic kinase whose activity is essential for mitotic entry and progression [65,66]. The phosphorylation of Arpp19 and ENSA by Gwl transforms these two proteins into potent inhibitors of PP2A-B55, the phosphatase identified as responsible for counterbalancing the pivotal mitotic kinase cyclin B/Cdk1 [63,64]. Gwl-mediated phosphorylation of Arpp19 and ENSA is performed at the FDSGDY sequence, a motif that is conserved in these two proteins from yeast to plants [67,68]. Data first demonstrated that this phosphorylation is essential and sufficient to promote Arpp19/ENSA-dependent PP2A-B55 inhibition and mitotic entry in interphase Xenopus egg extracts [63,64]. In the same line, the depletion of Arpp19 from mitotic Xenopus egg extracts results in mitotic exit by the re-activation of PP2A-B55 and cyclin B/Cdk1 substrate dephosphorylation [63]. A key role of these two proteins in conferring the correct temporal pattern of mitotic substrate dephosphorylation by PP2A-B55 has been reported in human and mouse cells. Accordingly, the knockdown/knockout of ENSA and Arpp19 results in dramatic mitotic defects including premature DNA decondensation and reformation of the nuclear envelope, as well as perturbed cytokinesis furrow formation [69,70,71].

Besides mitosis, Arpp19 and ENSA also control meiotic division. In this line, Arpp19 phosphorylation by PKA has been shown to be essential for prophase I arrest in Xenopus oocytes [72], whereas ENSA is required for prophase I exit in mouse oocytes [73].

Although both Arpp19 and ENSA regulate protein phosphorylation during mitosis and meiosis, they appear to perform differential roles in the control of the cell cycle of human and mouse somatic cells. This hypothesis is supported by previous results demonstrating that, whereas Arpp19 is essential for mice embryonic development by controlling mitotic division, ENSA is dispensable for mitosis and is mostly involved in normal DNA replication [70]. Accordingly, the knockdown of ENSA in human cells results in an extension of the timing of S phase consequent to the decline of the number of activated replication origins. This phenotype results from the dephosphorylation and subsequent ubiquitination and degradation of Treslin, a replication factor whose loading is essential for replication origin firing [74].

Interestingly, a new role of Gwl/Arpp19-ENSA/PP2A-B55 has also been suggested in platelet aggregation. In this model, the phosphorylation of ENSA and Arpp19 at the conserved FDSGDY motif has been demonstrated [75]. Moreover, a substitution of aspartic for glutamic acid at position 167 of the Gwl kinase was present in all the members of a thrombocytopenic family [76]. This mutation was associated with aberrant activation and survival of platelets, a phenotype that was accompanied by abnormal protein phosphorylation in these thrombocytes [77]. Further studies are required to establish the biological significance of the Gwl/Arpp19-ENSA/PP2A-B55 pathway in platelets and its impact on thrombocytopenia.

## 6. Signaling Cascades Regulated by PP2A-B55

PP2A-B55 regulates the activity of key proteins controlling cell growth and survival. These proteins include Akt, Ras/Mapk, Src, mTOR, and Wnt/βCatenin. 

The Akt pathway is activated in response to stimuli such as insulin, growth factors, or cytokines. This activation results in the phosphorylation of two different residues of Akt, Thr308 and S473, and potentiates the pro-proliferative and pro-survival properties of this kinase [78]. PP2A-B55 holoenzyme directly dephosphorylates Thr308 and inactivates Akt, resulting in the inhibition of cell growth and survival [79].

PP2A-B55 additionally participates in both the negative and the positive modulation of the Ras/Mapk pathway by dephosphorylating three key constituents of this cascade: The Kinase Suppressor of Ras 1 scaffold protein (KSR1), Raf1, and Mekk3. In response to growth factors, PP2A-B55 binds KSR1-Raf1 complex and dephosphorylates critical 14.3.3 binding sites on these proteins. Dephosphorylation of the KSR1-Raf complex promotes its dissociation from 14.3.3 and its recruitment to the plasma membrane resulting in Mapk activation [80]. Furthermore, PP2A-B55 holoenzyme can directly dephosphorylate an activatory site of Mekk3, causing its inhibition and thus participating in the negative regulation of this cascade [81]. A negative regulation of Mapk upstream of Raf1 has also been described for PP2A-B55 by directly dephosphorylating and inactivating Src [82]. 

PP2A-B55 has also been shown to positively modulate the Wnt/βCatenin pathway by directly dephosphorylating and preventing the ubiquitination and degradation of βCatenin [83]. This cascade plays an essential role in embryogenesis, however its misregulation in somatic cells results in tumorigenesis [84]. By maintaining βCatenin levels in somatic cells, PP2A-B55 can participate in the tumoral activity of this pathway.

Finally, PP2A-B55 additionally participates to the control of autophagy. Autophagy is an essential property used by cancer cells to survive under high energy consumption. This property is negatively regulated by the mTOR kinase that, by phosphorylating Ulk1, prevents its subsequent activatory phosphorylation by AMPK and the autophagic response. PP2A-B55 plays a prominent role in the control of autophagy via Ulk1 dephosphorylation by indirectly modulating mTOR activity and by directly dephosphorylating mTOR-dependent phospho-residues of this protein [85].

Besides Ulk1 phosphorylation, the autophagic response is regulated by the levels of the hypoxia-inducible factors (HIFs) as well. Under hypoxic conditions, these transcription factors promote the expression of Redd1 and the Redd1-dependent dissociation of TSC1/TSC2 from 14.3.3, permitting the binding of these inhibitors to mTOR and the negative modulation of this kinase. HIF stability is controlled by the hypoxia-inducible factor prolyl hydroxylase 1/2 (PHD1/2) [86]. These enzymes hydroxylate HIFs and target them for ubiquitination and degradation. PP2A-B55, by dephosphorylating and inactivating PHD1/2, stabilizes HIFs and triggers TSC1/TSC2-dependent mTOR inhibition, potentiating autophagy-mediated cell survival [87]. 

## 7. PP2A-B55 Misregulation in Cancer

PP2A was first suggested to act as a tumor suppressor based on the studies showing that okadaic acid, a chemical inhibitor of PP2A, caused tumors in mice [88]. This hypothesis was further supported by the fact that viral particles including the polyoma small and middle T antigens, the simian virus SV40 small T antigen, and the E4orf4 antigen promote cell transformation by inducing B55 subunit dissociation from PP2A [89,90]. Genetic and epigenetic alterations linked to B55 downregulation also support the role of PP2A-B55 as a tumor suppressor. Deletions and mutations on the B55 genes *PPP2R2* have been found in breast, prostate, primary plasma leukemia, acute myeloid leukemia (AML), and ovarian cell carcinoma. A study of 995 primary breast tumors identified heterozygous and homozygous deletions of the *PPP2R2A* gene associated with loss of B55α transcript in Estrogen Receptor (ER)-positive luminal B breast cancer [91] (Table 1). Somatic deletions were also observed in 67% prostate tumor samples [92] and deletions encompassing B55α and B55β genes *PPP2R2A* and *PPP2R2B* were found in primary plasma cell leukemia [93] and in multiple myeloma patients [94], respectively. Loss of function mutations on the *PPP2R2A* gene were additionally observed in acute leukemia blasts and were associated with the disappearance of B55α expression [95]. Interestingly, these cells were more sensitive to treatment with the Akt inhibitor, but less responsive to the PP2A activator FTY720, supporting the role of PP2A-B55α holoenzyme in the regulation of Akt phosphorylation in AML. 

Besides deletions and mutations, diminished protein expression of the B55 isoforms was found to be associated with different tumors. Accordingly, in a study of 511 AML patients, the expression of B55α was found decreased in AML cells compared to normal cells. This expression negatively correlated with Thr308 phosphorylation of Akt, supporting again the role of PP2A-B55α holoenzyme in Akt hyperphosphorylation in AML [96]. In the same line, decreased *PPP2R2A* mRNA has been found in lung and thyroid carcinomas [97] and downregulation of *PPP2R2B* by epigenetic silencing has been observed in colorectal cancer [98] and in ductal breast carcinoma [99]. The cellular levels of B55γ were analyzed by immunohistochemistry as well and found to be lost in prostate cancer, a fact that correlated with metastasis and mortality [100]. Finally, the expression of B55δ in hepatocellular carcinomas decreases inversely to the level of the 133b miRNA, a microRNA known to target B55δ mRNA. This suggests that 133b miRNA could act as an oncogene by reducing B55δ levels and PP2A-B55δ activity [101].

Although PP2A has been assumed to play a major role as a tumor suppressor, several studies indicate that this phosphatase can also activate several oncogenic cascades and positively participate in the oncogenic process [80,102]. Consistently, a significant enhancement in the levels of the B55α subunit has been described in pancreatic ductal adenocarcinoma. In these cancer cells, the increase of B55α levels correlate with poor survival of pancreatic cancer patients. Moreover, the depletion of this protein by siRNA in pancreatic cancer cell lines resulted in a drop of the phosphorylation of Akt and Erk1/2 and in a decrease of the levels of βCatenin. This phenotype was associated with a reduction of the oncogenic properties including attenuated cell growth, clonogenicity, mobility and anchorage-independent growth, supporting an effect of PP2A-B55α as an oncogene in pancreatic cancer [103]. 

## 8. Arpp19 and ENSA Misregulation in Cancer

Arpp19 overexpression associated with a gain of tumorigenic properties has been reported in a high number of cancers, indicating that, as expected by its PP2A-B55 inhibitory activity, it acts as an oncogene. Interestingly, Arpp19 upregulation in most cases results from the decreased levels of different microRNAs targeting this gene. 

In cervical cancer, the overexpression of the long noncoding RNA DLX6-AS1 results in the drop of the microRNA miR-16-5p and the upregulation of its downstream target Arpp19. DLX6-AS1 silencing attenuates oncogenic properties in cervical cancer cells, whereas its overexpression results in the opposite phenotype [104] (Table 2). 

In gastric cancer, other long coding RNA, IGFL2-AS1, is overexpressed. IGFL2-AS1 sponges miR-802, which in turn targets Arpp19 [105]. This alteration is associated with Arpp19 overexpression and increased oncogenic properties. miR-802 level attenuation has also been found in laryngeal cancer [106]. In these cancer cells, colony formation and cell proliferation, migration, and invasion are attenuated upon miR-802 ectopic expression and this attenuation is reversed by the overexpression of Arpp19. 

Two additional microRNAs, miR-26A and miR-320, have been shown to be involved in Arpp19 overexpression and cancer treatment resistance. miR-26A down-regulation in Papillary Thyroid Carcinoma is associated with increased cell proliferation and treatment resistance, and the knockdown of Arpp19 sensitizes these cancer cells to tamoxifen [107]. Similarly, downregulation of miR-320 in estrogen receptor positive breast cancer cells results in Arpp19 overexpression and in the loss of tamoxifen sensitivity, a phenotype that is reversed by miR-320 restoration [108]. 

Finally, increased Arpp19 mRNA expression has been reported in hepatocellular carcinoma [109] and AML [110] tissue samples, and high levels of this protein have been identified as an independent predictor of relapse in AML patients.

Unlike Arpp19, ENSA expression appears to be less involved in cancer development and progression. Surprisingly, although ENSA amplification and upregulation is observed in tumoral samples (cBioportal database), it has not been strongly linked with cancer patient outcomes, with only one study reporting the hypomethylation of the ENSA gene promoter and the overexpression of this protein in liver cancer cell lines and patient samples. Contrary to Arpp19, this overexpression was linked with tumor suppression [111].

Unfortunately, despite compelling data identifying Arpp19 as a potent oncogene, nothing is known about the underlying molecular mechanisms involved in oncogenesis and tumoral progression upon Arpp19 overexpression. No data exists about the putative involvement of Arpp19-dependent inhibition of PP2A-B55. Moreover, the direct targets of Arpp19 in the oncogenic process are completely unknown. Finally, nothing is known about the mechanisms by which ENSA, the other inhibitor of PP2A-B55, participates in the oncogenic processes. Further studies are required to uncover the exact role of these two proteins in cancer development and progression, the participation of PP2A-B55 in this phenotype, and to unravel whether they could represent interesting targets for cancer therapy.

## 9. Gwl/Mastl Misregulation in Cancer

As for Arpp19, Gwl (Mastl in humans) acts as an oncogene. The overexpression of Gwl has been reported in a high number of cancer types including hepatocarcinoma [112], neuroblastoma [113], head and neck squamous cell carcinoma [114], and breast [115,116,117,118], gastric [119], and colorectal [120,121] cancers. In all these cancers, the upregulation of Gwl correlates with increased risk of relapse and poor prognosis, specifically in head and neck carcinoma [114], neuroblastoma [113] and in breast [115,116,117], gastric [119], and colorectal [120] cancers. In these cases, the levels of this kinase could be used as a valuable indicator of metastatic relapse and death. 

The upregulation of Gwl is also associated with an increase of the oncogenic capacities such as cell proliferation, migration, and invasion [114,116,120,121], and its silencing is linked with a drop of these properties as well as with a drop of xenograft tumor growth “in vivo” [115,116,120,121,122]. Decreased cell proliferation under these conditions is mostly induced by mitotic catastrophe and apoptosis [115,116,122]. 

Interestingly, all tumoral cells are not impacted at the same level by Gwl knockdown. Notably, whereas some breast cancer cell lines such as MDA-MB-231 or BT-549 are highly sensitive to Gwl knockdown, others such as EVSA-T or T47D are largely insensitive [115]. For some cancer cells, including AML cells [123] and thyroid cancer cells [124], Gwl depletion was specifically affecting tumoral but not normal cells, pointing to Gwl as a novel therapeutic target. In the same line, the downregulation of this kinase has been shown to sensitize tumoral cell response to anticancer treatments such as radiation in breast [122] and non-small cell lung cancer cells [125], cisplatin in thyroid cancer cells [126], and 5 fluorouracil in colorectal cancer cells [120].

The oncogenic activity of Gwl has been attributed to the modulation of different underlying signaling cascades. First, analysis in breast and colorectal cancer cell lines pointed to a role of Gwl overexpression in the hyperphosphorylation and hyperactivation of the oncogenic kinase Akt. The reported data demonstrated that increased Gwl levels correlates with dephosphorylation and activation of the GSK3 kinase as well as with the phosphorylation and subsequent SCF-dependent ubiquitination and degradation of PHLPP, the phosphatase targeting Akt. A model was proposed in which Gwl, via GSK3 activation, promotes PHLPP degradation and as a consequence, Akt hyperphosphorylation and hyperactivation, thus potentiating its oncogenic activities [121].

In a second study, a role of Gwl in the upregulation of the Wnt/βCatenin pathway and of the anti-apoptotic proteins Survivin and Bcl-xL has also been suggested. Notably, the levels of these antiapoptotic proteins correlates with Gwl levels in HCT116 and SW620 colorectal cancer cells. Gwl overexpression was associated with an increase of the amount of βCatenin and of its target c-Myc that was attributed to the phosphorylation and inactivation of GSK3 [120]. From this data, authors proposed that Gwl silencing could positively modulate GSK3 dephosphorylation and activation, and destabilization of βCatenin. Low βCatenin levels would result in decreased expression of cMyc and of its targets Survivin and Bcl-xL, and would promote colorectal cancer cell sensitization to 5-fluorouracil treatment. 

Finally, Gwl overexpression has been proposed to play a pleiotropic role via the drop of PP2A-B55-dependent dephosphorylation of multiple components of the different signaling pathways. These pathways regulate actin dynamics, cell-cell junctions, MAPK signaling, and DNA damage [116]. A SILAC-phosphoproteomic analysis in MCF10 breast cancer cells overexpressing the Gwl kinase revealed an enrichment of up or down phosphorylated proteins upon Gwl overexpression. The up-regulated phosphopeptide group upon Gwl overexpression displays an enrichment of proteins involved in desmosomes, actin dynamics, MAPK signaling, and DNA damage. Conversely, down-phosphorylated proteins were clustered around protein kinase D1, vimentin, and Akt downstream targets. According to a role of Gwl in the control of cell-cell junctions, the overexpression of this kinase in MCF10A cells resulted in the disorganization of actin cytoskeleton and the reduced expression and mislocalization of E-cadherin and βCatenin. As a consequence, Gwl overexpression disrupted collective cell migration [116]. Interestingly, a recent report identified Gwl as an activator of cell contractility via the positive modulation of MRTF-A/SRF (myocardin-related transcription factor A/serum response factor) signaling [127]. By directly binding to MRTF-A, the Gwl kinase promotes the nuclear retention of this transcription factor and mediates the transcription of genes related to actin dynamics, cell adhesion, and actomyosin contraction via its association to SRF. Altogether, these data suggest a pleiotropic role of Gwl on the control of different signaling pathways involved, not only in cell cycle control, but also in cell proliferation, cell survival, and cell-cell contact regulation. Unfortunately, although the causes of the oncogenic Gwl activity are beginning to be understood, nothing is known about the underlying mechanisms promoting Gwl overexpression and tumor formation.

## 10. Summary

Phosphatases have been identified as tumor suppressors very commonly misregulated during cancer and consequently have become suitable candidates for anticancer therapies. From all these phosphatases, PP2A-B55 is emerging as a key holoenzyme often misregulated in a high number of cancer types. Although the constituents and the protein structure of this complex have been largely investigated, the molecular mechanisms regulating its activity were unknown, notably, no specific inhibitors of this holoenzyme were identified. The discovery of the Gwl/Arpp19-ENSA pathway, together with the finding of an increasing number of PP2A-B55 substrates and the advances on the molecular mechanisms conferring substrate specificity, bring to the forefront the study of this holoenzyme, both under physiological and pathological conditions. In this review, our objective was to summarize the main data recently provided in the bibliography about PP2A-B55 and to highlight the prominent impact of its misregulation in the oncogenic process.

## Figures and Tables

**Figure 1 biomolecules-10-01586-f001:**
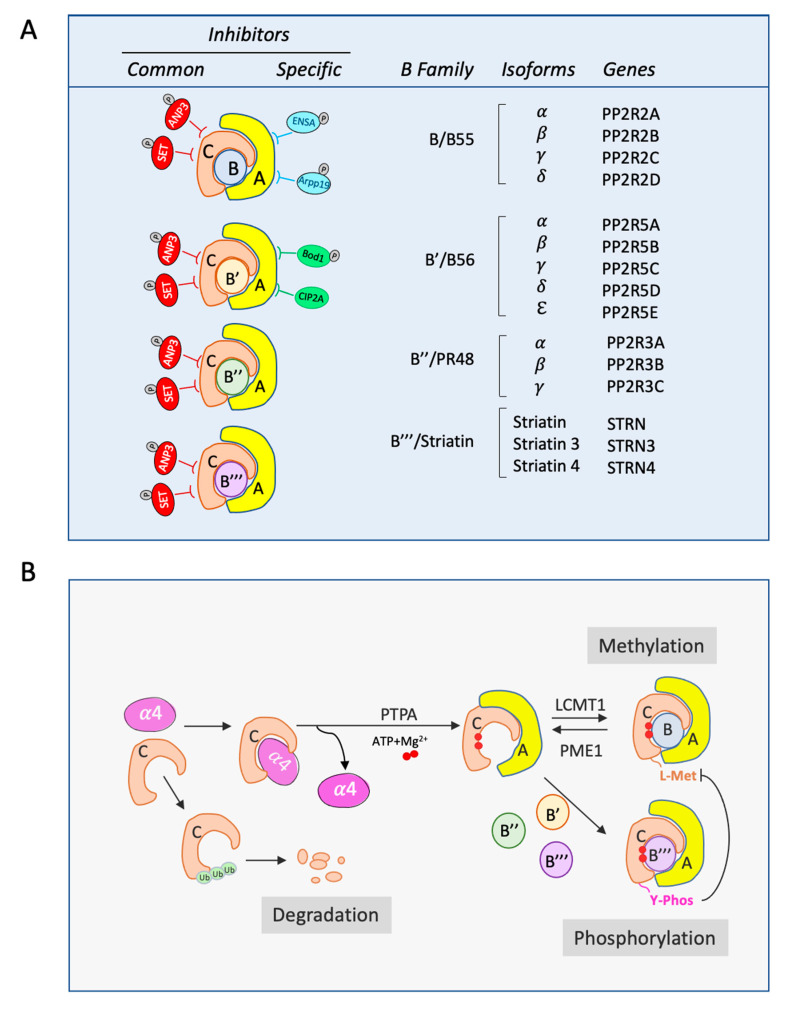
Composition and biogenesis of the PP2A holoenzymes. (**A**) Genes coding for the different isoforms of PP2A B subunits. Represented are also common and specific inhibitors of the PP2A holoenzymes. (**B**) Biogenesis of the PP2A complex. Monomeric C subunits binds α4 protein or is directly ubiquitinated and degraded. C/α4 complex formation mediates A/C association that will be subsequently activated by PTPA by the introduction of Mg^2+^ ions (red circles). This activation will permit the methylation or phosphorylation of the C subunit and the formation of PP2A-B55 or of the other holoenzymes respectively.

**Table 1 biomolecules-10-01586-t001:** Alterations reported for the distinct B55 isoforms in different types of cancers.

**PP2A-B55: Tumour Suppressor Activity**				
**Subunit**	**Gene**	**Alteration**	**Disease**	**Ref.**
B55𝛼	*PPP2R2A*	Deletions	Luminal B breast cancer	[91]
B55𝛼	*PPP2R2A*	Deletions	Prostate cancer	[92]
B55𝛼	*PPP2R2A*	Deletions	Primary plasma cell leukaemia	[93]
B55𝛽	*PPP2R2B*	Deletions	Myeloma	[94]
B55𝛼	*PPP2R2A*	Loss of function mutation	AML	[95]
B55𝛼	*PPP2R2A*	B55 downregulation	AML	[96]
B55𝛼	*PPP2R2A*	Decreased mRNA	Lung & thyroid carcinoma	[97]
B55𝛽	*PPP2R2B*	Epigenetic silencing	Colorectal cancer	[98]
B55𝛽	*PPP2R2B*	Epigenetic silencing	Ductal breast carcinoma	[99]
B55𝛾	*PPP2R2C*	B55 downregulation	Prostate cancer	[100]
B55𝛿	*PPP2R2D*	Up miRNA/down mRNA	Hepatocellular carcinoma	[101]
**PP2A-B55: Oncogenic Activity**				
**Subunit**	**Gene**	**Alteration**	**Disease**	**Ref.**
B55𝛼	*PPP2R2A*	B55 overexpression	Pancreatic cancer	[103]

**Table 2 biomolecules-10-01586-t002:** Mechanisms triggering Arpp19 overexpression in the indicated types of cancers.

Arpp19 Overexpression			
Alteration	miRNA	Disease	Ref.
*Long Noncoding RNA DLX6-AS1 overexpression*	Drop miR-16-5p	Cervical cancer	[104]
*Long Noncoding RNA IGFL2-AS1 overexpression*	Drop miR-802	Gastric cancer	[105]
*_*	Drop miR-802	Laryngeal cancer	[106]
*_*	Drop miR-26A	Papillary thyroid Cancer	[107]
*_*	Drop miR-320	Breast cancer	[108]
*Increased mRNA levels*	_	Hepatocellular carcinoma	[109]
*Increased mRNA levels*	_	AML	[110]

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
