# Peer review of "PP2A-B55 Holoenzyme Regulation and Cancer"

_biomolecules, 2020, doi:10.3390/biom10111586_

Round 1

Reviewer 1 Report

The authors have provided a general comprehensive review of the role of PP2A-B55 in cell proliferation and cancer. The review will help to synthesize new concepts and poses questions that need to b studied. A few minor comments on the text are detailed below: 

1) The authors give the impression that the recent discovery of inhibitors of the PP2A-B55 holoenzyme was the impetus for this review. They should articulate whether such reviews exist and their intention to fill any gaps as the rationale for this review. 

2) There are several statements in the manuscript that require editing for clarity and language. 

3) For example: Page 2: Lines 52-53. 

Author Response

We thank this referee for their comments.

1) In this new version of the manuscript, the first concern of this referee has been addressed in lanes 53 to 57 where we pointed out that, despite the high number of reviews appeared in the last years on the function of Gwl/Arpp19-ENSA/PP2A-B55 in cell cycle regulation, only few (2 reviews) were focused on the impact of the misregulation of this pathway in cancer. Our main aim is to fill this gap in the bibliography.

2) and 3) The manuscript has been edited and the different unclear statements corrected.

Reviewer 2 Report

This is a useful review by very knowledgeable experts who have made important contributions to our understanding of PP2A-B55 regulation. It should be published. I have only a few comments.

-In some places, I felt that it was premature to give so much space in a review to studies that have left the mechanisms unclear. In particular, the only figure of the review shows contradictory models proposed by two studies on the effect of Gwl overexpression in cancer, where in both cases, the substrate of Gwl involved was not even determined.

-I suggest adding a figure to illustrate the different PP2A holoenzymes and their regulation by inhibitors, activators and post-translational modifications.

-An additional effort should be made to rely on and cite primary literature instead of reviews. For example, saying that methylation of PP2A C by LCMT-1 impacts the formation of PP2A-B55 only is an oversimplification in view of the published literature (line 126).

Author Response

We thank the referee for their suggestions that will help us to balance the different parts of the manuscript and to improve their quality.

Following his/her suggestion, Figure 1 of the manuscript has been substituted for a new figure depicting the different B subfamilies, isoforms and the specific inhibitors of PP2A, as well as the biogenesis of these holoenzymes.

In addition, the different studies describing the precise role of LCMT1 on PP2A holoenzyme formation have been reported in lines 130-137 and the corresponding citation introduced.

Reviewer 3 Report

The review on B55 by Goguet-Rubio et al is very well written and structured. It provides a comprehensive survey on the regulation of B55 and its implication in cancer. It also covers the regulatory elements upstream of B55, e.g. Gwl, and how this misregulation could be casually linked to certain cancer types. This reviewer recommends publication of the review as it is following correction of these minor points:

- line 72: scaffold instead of scalfold

- line 72: please explain the abbreviation HEAT repeates the first time it appears

- line 217: please specify to what “its” refers to

- line 255: please explain the abbreviation “ER-positive”

Author Response

The different minor points of this referee have been corrected in the text.